# Larval shell chemistry of the Olympia oyster *(Ostrea lurida)* in Puget Sound, WA to assess population connectivity and restoration planning

**Megan Hintz**[1], **Bonnie J. Becker**[2]*, **Henry S. Carson**[1], **Verena H. Wang**[3], **Marco B. A. Hatch**[4], **Brian Allen**[5], **Brian Rusk**[4]

**1** Washington Department of Fish and Wildlife, Olympia, Washington, United States of America, **2** University of Washington Tacoma, Tacoma, Washington, United States of America, **3** Gulf of Mexico Fishery Management Council, Tampa, Florida, United States of America, **4** Western Washington University, Bellingham, Washington, United States of America, **5** Puget Sound Restoration Fund, Bainbridge Island, Washington, United States of America

\* bjbecker@uw.edu

## Abstract

The Olympia oyster (*Ostrea lurida*) is the only native oyster species along the west coast of North America and is culturally and ecologically important. However, Olympia oyster populations have been severely depleted, prompting ongoing restoration efforts in Puget Sound, WA, and beyond. Understanding population connectivity is vital for successful restoration planning to ensure resilience and genetic diversity. This study examined the potential for using trace elemental "fingerprints" in Olympia oyster shells to track larval dispersal and connectivity at regional scales within Puget Sound. Brooded larvae were collected via non-lethal sampling at eight sites grouped into three geographic regions. Shell chemistry analysis showed the ability to distinguish these regions from each other with approximately 75% accuracy, demonstrating feasibility for addressing connectivity questions among sub-basins. Additionally, regional signatures were found to be temporally stable within one reproductive season, facilitating annual sampling regimes. Although settlers of unknown origin collected at two restoration sites could not yet be confidently assigned to specific source regions, nor could they be divided into groups in a cluster analysis, likely due to methodological constraints, this study provides a proof of concept and foundation for further developing this technique. With targeted improvements to analytical methods for microscopic larval shells, shell elemental fingerprinting shows promise to greatly inform ongoing restoration efforts by elucidating population connectivity patterns for this culturally and ecologically important native oyster species at ecologically relevant scales.

## Introduction

Ensuring population connectivity, the exchange of individuals among geographically separated subpopulations, is imperative to successfully managing marine species. For sedentary species, dispersal of the planktonic larval phase is crucial to population connectivity.

**Data availability statement:** All relevant data are within the manuscript and its Supporting Information files.

**Funding:** This study was funded by Washington Sea Grant (R/HCE-8, BJB), https://wsg.washington.edu/. The funders had no role in study design, data collection and analysis, decision to publish, or preparation of the manuscript.

**Competing interests:** The authors have declared that no competing interests exist.

Planktonic larvae can travel vast distances, and for sedentary species, this is the primary opportunity to disperse to neighboring subpopulations. Metapopulation dynamics comprising the level of connectivity and significance of self-recruitment inform spatial management of many marine invertebrates [1]. Larval connectivity is an important factor to consider in the conservation, restoration, and management of marine species that can help determine the size, configuration, and location of reserves and restoration efforts with the ultimate goal of establishing self-sustaining populations [2,3,4,5]. Quantifying larval connectivity dynamics is complex and remains difficult to determine due to small larval size, larval behavior, high mortality, and large dispersal potential. For this reason, larval connectivity dynamics are often poorly understood despite their high importance.

One species that would benefit greatly from a better understanding of larval connectivity is the Olympia oyster, *Ostrea lurida* [6]. As the only native oyster found between Alaska and Baja, Mexico, they hold cultural, ecological, and economic significance throughout their range. Olympia oysters were reduced to near extinction over 100 years ago, and less than 5% of their historical abundance remains in Washington's Puget Sound today [7,8,9,10]. Efforts to restore the native Olympia oyster to Puget Sound have been ongoing for over 20 years and have successfully increased their spatial extent and abundance [11]. Restoration efforts would benefit from the knowledge of source-sink dynamics, larval dispersal distances, and the level of population connectivity to establish a network of self-sustaining subpopulations.

Natural geochemical tags incorporated into calcified structures of marine organisms can record individual movement. Geochemical tags are generated by physical and chemical variations in environmental conditions (e.g., temperature, salinity, and abundance of certain other elements) and can be used to directly track newly settled individuals back to their natal location [12]. With the ability to track individual recruits, population connectivity can be quantified on ecologically relevant time scales that are important for informing restoration. Genetic approaches to evaluating population connectivity estimate connectivity over evolutionary time scales [13] and the exchange of only a few individuals, whether via natural dispersal or assisted by anthropogenic transfers, can maintain genetic homogeneity among subpopulations [14]. To maintain ecologically relevant larval exchange, immigration and emigration between subpopulations need to be several orders of magnitude higher than what is required to maintain genetic homogeneity [15]. One important question in restoration science is, to what extent does a restored or enhanced sub-population augment neighboring sub-populations with immigrants and/or rely on emigrants from neighboring sub-populations? Trace elemental fingerprinting provides a mechanism to answer these questions by determining the natal origin of individual recruits.

Olympia oyster shells incorporate elements from seawater into their shell in proportion to environmental concentration, temperature, and salinity [16], providing tags to track larval movements [17]. Because their larvae are brooded for the first phase of development, the pelagic larval duration and dispersal potential of *O. lurida* is likely shorter than oysters that do not brood. The actual pelagic larval duration is unknown, but is as short as 7 days in laboratory trials [18]. Elemental fingerprinting has been applied to shelled molluscs to distinguish between estuaries along the coast [17,19–21] and small scale within small estuaries (1–10 km) [22,23], but has never been used to distinguish locations within a large estuarine system such as Puget Sound for invertebrates. Within Puget Sound, elemental signatures have been applied to multiple fish species to determine the maternal origin of salmon offspring [24], distinguish between estuarine and ocean natal sources [25], identify nursery habitats [26], and improve the understanding of population structure [27] of fish utilizing otolith chemistry. Significant spatial variability in trace elements of otoliths allowed for successful classification among regions in Puget Sound [27] with fine-scale resolution identifying regions with as little as 10 km separation [26].

The goals of this study were to:

(1) Examine the spatial and temporal variation of trace elemental signatures among broods to explore the use of fingerprinting for marine invertebrates in Puget Sound for the first time,

(2) Assign unknown-origin settlers to one of the source populations by matching the chemistry of the brooded portion of their shell to collected broods of known origin, to assess population connectivity to and from restored populations, and

(3) Test the effect of a non-lethal sampling of brooding oysters on the resulting trace elemental values in the larval shells.

## Materials and methods

### Study location

Puget Sound is a large glacially carved estuarine system in the northwest corner of the continental United States and forms the southern portion of the Salish Sea. As one of the world's most productive nearshore bodies of water, Puget Sound is home to a diverse and economically important ecosystem. This complex estuarine system is an extension of the Strait of Juan de Fuca with over 3,000 kilometers of intricate shoreline with frequent freshwater inputs surrounding numerous islands of all sizes [28]. Water flow within Puget Sound and among its five geographically defined sub-basins is dominated by the strong tidal influence [29], leading to a mean residence time of 57 days within these regions [30]. The diversity of land use, geology, and freshwater inflow throughout the estuary potentially provide high variability of elemental signatures spatially [10].

### Field sampling

To characterize unique elemental signatures within the Puget Sound region, 12 sites with populations of *Ostrea lurida* were searched for the presence of brooded larvae (Fig 1). Potential collection sites were chosen using existing information about adult oyster populations, prioritizing populations of substantial size while encompassing a large geographic range. We did not divide the sites into "regions" prior to trace elemental analysis of the larval shells. Regions were defined post-hoc using larval shell chemistry. All field collections for *O. lurida* adults, brooded larvae, and later for post-settlement recruits on public lands were authorized by a Scientific Collection Permit issued by the Washington Department of Fish & Wildlife; on private lands, permissions for access and collection were provided by the owner. Each site was visited from June 15th to Aug 15th, 2015, and 51–578 adults per site were sampled to assess the reproductive status of individual *O. lurida* and maximize the number of brooded larvae collected. Two locations sampled, Fidalgo Bay and Dyes Inlet, were sampled biweekly throughout the summer to assess the temporal variation of elemental signatures. Eleven other sites, including a second site in Dyes Inlet, were sampled 1–4 times to maximize the locations of brooded larvae and the number of brooded larval samples collected. Brooded larvae were successfully collected from 9 of the 12 locations in Puget Sound sampled (Fig 1, Table A in S1 Appendix).

Adult oysters were collected non-destructively on-site and anesthetized using $MgSO_4$ [31]. Treated individuals opened their valves and were visually inspected for brooding larvae. Early-stage trochophore larvae identifiable by their light color were noted and discarded. Darker late D-stage veliger larvae that had formed their initial calcified shell, the prodissoconch, were collected. These late-stage brooded larvae were carefully rinsed from the mantle cavity with filtered seawater (75 microns collected at location). All larvae were rinsed in three

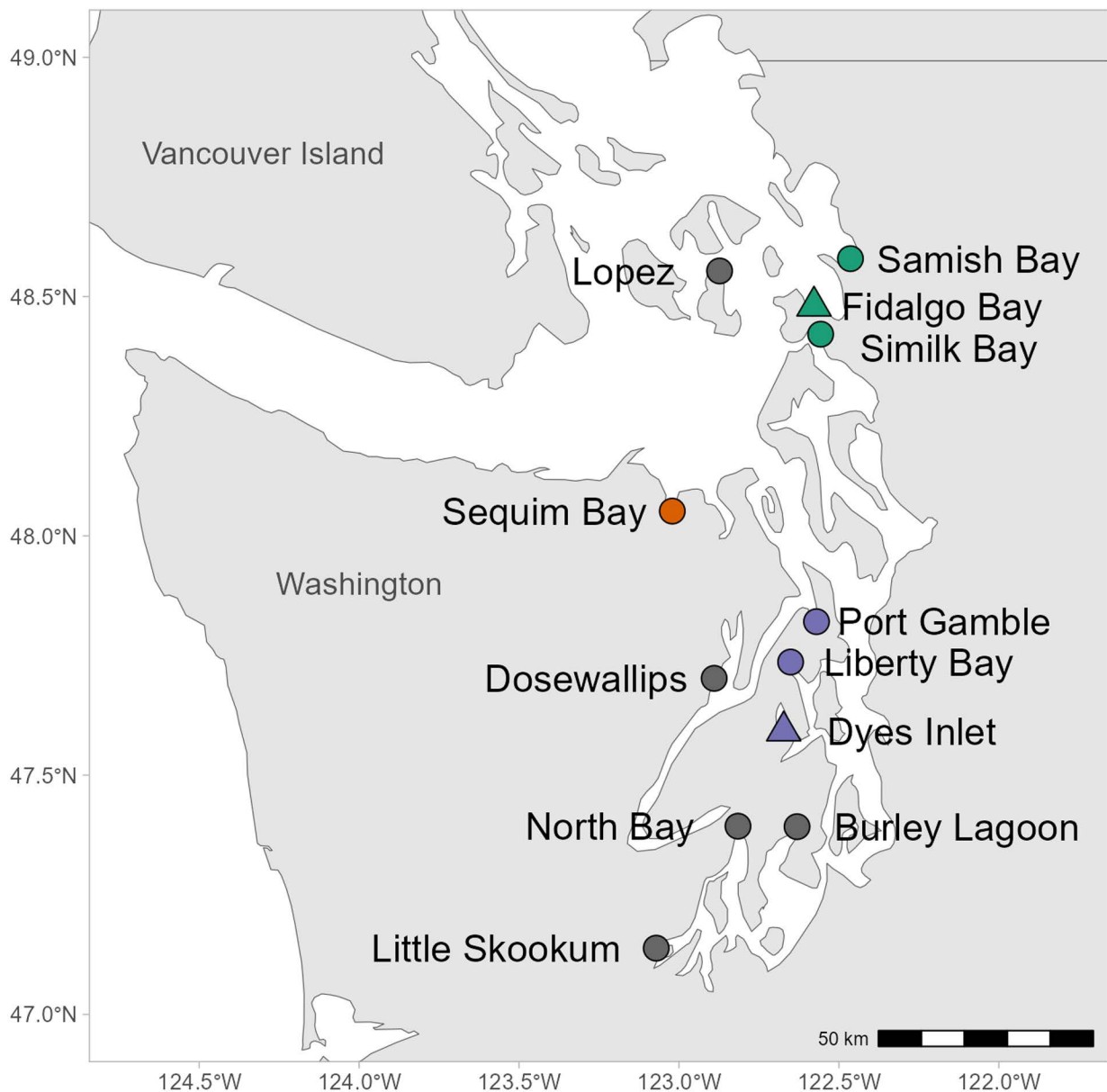

**Fig 1. Map of Puget Sound, WA, USA, representing sites of existing *Ostrea lurida* populations that were sampled for brooded larvae.** Symbol color represents the regions that were defined post-hoc by their differing elemental signatures. Triangle icons mark the two sites that were sampled more frequently. Map made from Natural Earth. Free vector and raster map data @ naturalearthdata.com.

different solutions each with decreased salinity, filtered seawater, 50:50, and ultrapure water to remove any remaining anesthetic solution in the sample before being stored in ultrapure water. Larval samples were stored in two 1.5-mL acid-leached and rinsed Eppendorf tubes per brood and frozen upon return to the laboratory. Late-stage brooded larval samples were found and collected from 99 oysters spanning nine different locations in Puget Sound (Table A in S1 Appendix).

To limit trace element contamination of samples, all containers, equipment, and other materials coming in contact with larval samples in the field were acid-leached in 0.1 mol/L

hydrochloric acid overnight, and sample storage vials were acid-leached in 1.0 mol/L hydrochloric acid at sub-boiling temperatures for 8 hours, then rinsed five times with ultrapure water (resistivity > 18.2, Millipore).

*Ostrea lurida* recruits, referred to as settlers from here on out, are defined as individual larvae that have gone through metamorphosis and attachment on settlement collectors during the period of deployment. The settlers were collected with tile settlement collectors concurrent with brooded larval collection. Tile settlement collectors were constructed with eleven 100 cm$^2$ porcelain tiles, rough side down, stacked between two polyvinyl chloride plastic (PVC) pipes that were driven into the sediment with the bottom tile flush to the sediment to mimic shell string collectors [19,32,33]. Three replicate tile settlement collectors were placed near adult *O. lurida* beds in Fidalgo Bay and Dyes Inlet and collected biweekly from 6/16/2015–8/27/2015. After deployment, the collectors were retrieved, the bottom tile was discarded, and tiles were stored in the freezer until further processing. A subset of settlers were collected from the tiles and analyzed in this study. Settlers ranged in size from 195 to 1755 microns in maximum diameter.

## Laboratory methods

**Sample preparation for LA-ICP-MS Analysis.** Sample preparation methods were modified following Carson [17] and Becker et al. [19] to clean and prepare early-stage and settler bivalves for trace elemental fingerprinting. To clean the shells and remove organic matter, isolating the elemental signatures of the inorganic component of the shell, shells were cleaned with 15% hydrogen peroxide ($H_2O_2$, UltraTrace, Fisher Scientific) buffered with 0.05 mol/L NaOH (Trace Metal Grade, Fisher Scientific). All containers, equipment, and other materials coming into contact with larval and settler shell samples in the lab were acid-leached in 1 mol/L hydrochloric acid for one week and then rinsed five times with ultrapure water (resistivity > 18.2, Millipore).

A few hundred brooded larval shells per parent oyster were carefully isolated and placed in 1 mL of buffered hydrogen peroxide cleaning solution overnight (12-18h) at room temperature. Shells were rinsed in ultrapure water three times before being transferred into a petri dish and visually inspected. Only clean larval shells were transferred into the final rinsing vial and rinsed with ultrapure water two more times, then carefully crushed to homogenize. The homogenized shell mixtures were dried on a clean glass slide before being transferred and mounted on 3M-brand Scotch double-sided tape affixed to glass slides. Larval samples were prepared and mounted in random order, and completed slides were stored in a laminar flow hood until elemental analysis.

Individual settlers collected from Fidalgo (n = 81) and Dyes Inlet (n = 73) collection tiles were cleaned and mounted following the same protocol as brooded larval shells with one exception. The top valve of individual settler shells was mounted directly on double-sided tape affixed to glass slides; settler shells were not homogenized.

**LA-ICP-MS analysis.** Shells were analyzed using a New Wave Research UP 213-nm laser attached to an Agilent 7500 series ICP-MS. Each glass slide was placed into a "super cell" low-volume sample chamber with helium used as the carrier gas. The chamber also contained the U.S. Geological Survey microanalytical carbonate standard number three (MACS-3), a matrix-matched standard. Sets of three ablation passes through the standards before, during, and after each slide were used to calibrate the ICP-MS and correct for instrument drift. Since brooded larval shells from each parent oyster were mounted as concentrated masses of shell mixtures, they produced a composite of the elemental signature of the sample, reducing the variability between individual larvae within a sample. Ablation passes for each piece of double-sided tape

were used to monitor for contamination, and none of the analyte elements were detectable in any analysis of the tape. Homogenized brooded larvae samples were ablated at a fluence of 2–3 joules per second and 10 Hz. One-millimeter-long line scans of homogenized shells were conducted using a 55-micron spot size and a raster rate of 30 microns per second. Three short lines were ablated on each settler shell, one for each part of the shell formed during brooding, dispersal, and after settlement following Carson [17], only information collected from the brooded ablation line was used in sourcing settlers to brood origin. These relatively weak settings were used to minimize burn-through of the shell into the mounting tape determined by visual inspection. Elemental values for each ablation line were calculated from raw values using Glitter Software.

Concentrations of 12 elements were quantified: $^{26}$Mg, $^{27}$Al, $^{31}$P, $^{42}$Ca, $^{55}$Mn, $^{56}$Fe, $^{63}$Cu, $^{67}$Zn, $^{69}$Ga, $^{88}$Sr, $^{137}$Ba, $^{208}$Pb, $^{238}$U. These elements were chosen because they are found in Puget Sound, WA waters or sediment or have previously been successful in elemental fingerprinting of bivalves [34] and were detected in larval shell samples in preliminary trials. Elemental concentrations for analysis were expressed as µmol mol$^{-1}$ ratios relative to $^{42}$Ca (element:Ca). Brood shell ablations for three elements (Cu:Ca, Ga:Ca, and Pb:Ca) consistently fell below limits of detection (LOD) and were removed from further analysis (Table B in S1 Appendix).

**Influence of non-lethal sampling method on elemental signatures.** To determine if the non-lethal sampling method of collecting brooding larvae by anesthetizing the adults in a $MgSO_4$ solution influenced the elemental signatures of larval shells, we compared the elemental concentrations in shells collected using the non-lethal $MgSO_4$ anesthetic to the lethal collection method of shucking the adult. During June 2016 in Fidalgo Bay, WA (48.477810°N, -122.574217°W), we collected oysters from fourteen haphazardly placed 1/16 m$^2$ quadrats in aggregations of *O. lurida*. All oysters collected were checked for reproductive status using either the $MgSO_4$ anesthetic or the lethal collection method. Late-stage brooded larval samples were collected from nine different oysters; four of which were collected using the $MgSO_4$ anesthetic, and five were lethally sampled. All brooded larvae samples were rinsed following the protocol used during collection in 2015. Elemental concentrations in these brooded larvae shells were determined following the same LA-ICP-MS methods of preparation and analysis outlined above for trace elements of $^{11}$B, $^{25}$Mg, $^{27}$Al, $^{29}$Si $^{31}$P, $^{55}$Mn, $^{56}$Fe, $^{63}$Cu, $^{67}$Zn, $^{88}$Sr, $^{137}$Ba, $^{208}$Pb, $^{238}$U as relative ratios to $^{42}$Ca.

## Statistical analysis

Final *O. lurida* elemental signatures consisted of nine trace elemental ratios (Mg:Ca, Al:Ca, P:Ca, Mn:Ca, Fe:Ca, Zn:Ca, Sr:Ca, Ba:Ca, U:Ca). The ten replicate ablations from each brooded larvae sample were averaged to create the best representation of the elemental signature for that brooded larvae sample. Before all multivariate analyses, elemental data were log-transformed and standardized (mean-centered, unit variance), and outliers were removed. All statistical analyses were performed in R (R version 3.4.2, RStudio version 1.1.383).

**Brood signature classification analysis.** The elemental signatures of larval samples from 99 individual Olympia oysters from 9 different locations of oysters in Puget Sound, WA, were used to discriminate between geographically distinct sites and regions using linear discriminant analysis (LDA). LDA predicts group membership by building discriminant axes that are linear combinations of elemental ratios by minimizing the within-group variance and maximizing the between-group variance [35]. Assignment accuracy was determined by leave-one-out cross-validation (jackknife), where the classification model was repeated with each sample sequentially withheld, and classification success was evaluated using the withheld sample. This method assumes multivariate normality and equal co-variances, which can often be met by transforming the data.

Selecting a combination of variables that maximizes accurate assignment to groups is vital to optimizing classification [36]. LDA was performed on all combinations of variables (9 variables, 511 combinations) to determine which combination of variables produced the model with the highest assignment accuracy. LDA was performed on log-transformed elemental ratios to meet assumptions of normality and equal co-variances to assess our ability to distinguish between regions based on shell chemistry. LDA was applied to individual sites as well as the sites grouped together into empirically defined geographic regions: North Sound (NS, which includes Samish Bay, Fidalgo Bay, and Similk Bay), Sequim Bay (Seq), and Central Sound (CS, which includes Port Gamble, Liberty Bay, and two sites in Dyes Inlet) (Fig 1, Table A in S1 Appendix). South Sound was excluded from the regional analysis due to low sample size in the only site at Little Skookum.

LDA analysis classification success was determined by the overall and site/regional cross-validated jackknifed assignment accuracy. However, with unequal group sizes, LDA jackknife may reclassify many samples accurately simply by chance [37]. To determine a more accurate estimate of classification success for the best-performing variable combination for LDA to region, a 90/10 random bootstrap sample was used. Randomly 90% of the data were subset to create the LDA classification model, and the assignment accuracy was determined by classifying the remaining 10% of the data. The data were subsampled 5,000 times to determine the mean assignment accuracy and confidence intervals. Additionally, elemental ratios used in the optimal LDA classification model were compared among regions utilizing univariate one-way analysis of variance (ANOVA, df = 2) followed by post hoc Tukey's tests using log-transformed (but not standardized) data.

**Temporal variation.** To examine the relative influence of temporal variation on all of the elemental ratios, non-parametric multivariate Analysis of Similarities (ANOSIM) procedures were conducted on Gower's similarity matrix and paired with Nonmetric Multidimensional Scaling (NMDS, two ordination axis and 100 random starts) on brooded larvae samples collected biweekly from two locations separately, Fidalgo Bay (North Sound) and Dyes Inlet (Central Sound). Samples collected from 6/15/2015–7/5/2015 were categorized as early season (n = 9 and 10 for Fidalgo Bay and Dyes Inlet, respectively), and samples collected from 7/6/2015–8/15/2015 were categorized as late season (n = 9 and 14 for Fidalgo Bay and Dyes Inlet, respectively) (Table A in S1 Appendix). All nine elemental ratios (Mg:Ca, Al:Ca, P:Ca, Mn:Ca, Fe:Ca, Zn:Ca, Sr:Ca, Ba:Ca, U:Ca) were used in this multivariate analysis. ANOSIM was conducted to test for significant differences between early and late seasons for each location sampled independently, and the model's goodness of fit was assessed by comparing the observed dissimilarity to the ordination distance.

**Influence of non-lethal sampling method on signatures.** To evaluate the potential influence of our non-lethal sampling method for collecting brooded larvae using a $MgSO_4$ solution, ANOSIM and NMDS (as described above) were conducted to determine whether there were significant differences between collection methods between anesthetized and lethally sampled broods. To further examine the influence of non-lethal sampling methods on elemental signatures, both anesthetized and lethally collected samples were plotted in brood signature LDA ordination space.

**Assignment of settler signatures to brood origins.** After verifying that *O. lurida* brood elemental signatures can accurately discriminate among broad geographic brood regions, the elemental profiles of brood regions are typically used as a reference map to retrospectively assign *O. lurida* settlers to brood regions of origin. Brood signature parameters can be used to estimate settler group membership either with LDA [38] or by applying a mixed stock algorithm to derive maximum likelihood estimates (MLE) [39] of settler sample mixture proportions and classify unknown settlers to brood origins. Both assignment methods require

that analytical sampling of trace elemental signatures is equivalent between brooded larvae and settler natal shell regions.

We first plotted settler elemental signatures to evaluate congruence in brood and settler elemental signatures in brood LDA ordination space. Both LDA and MLE methods assign all settlers to a brood source origin and cannot account for potential unsampled brood sources or settler signatures that are distinct from baseline source signatures. To further assess whether brood and settler signatures were comparable, we estimated the proportion of settlers unlikely to have originated from brood sources following the methods used in Standish et al. 2008 [40] (see S1 Appendix for detailed methodology). A large proportion of distinct settlers may be indicative of mismatched elemental sampling methodology between the two stages of *O. lurida* or the contribution of unsampled brood origin sites.

**Settler signature cluster analysis.** Because *O. lurida* settler signatures appear to be poorly represented by brood source regions, the LDA and MLE classification methods described above would likely result in inaccurate brood origin assignments. In such a case, where a reliable baseline reference map of brood origins is unavailable, estimating the number of geochemically defined brood sources contributing to settler populations can provide valuable information about larval dispersal and population connectivity. We used a Markov-chain Monte Carlo (MCMC) clustering algorithm [41] to identify the number of distinct clusters of elemental signatures (brood sources) present in the *O. lurida* settler cohort, estimate the contribution of each putative brood sources to the settler populations, and assign individual settlers to brood sources (see S1 Appendix for detailed methodology). This MCMC clustering method has been successfully applied in previous studies to estimate the number of distinct natal sources, defined by unique otolith geochemistry, contributing to fish populations [41,42,43].

## Results

### Brood signature classification results

Three regions were found to have distinct elemental fingerprints in Puget Sound (Fig 2). At the site level, overall assignment accuracy was low (42.4%) using an optimized elemental signature (Mg:Ca, P:Ca, Mn:Ca, Fe:Ca, Zn:Ca, Sr:Ca, U:Ca) to classify Olympia oysters (n = 99) to 9 sites. Oysters were more accurately assigned into broad regions (North Sound, Sequim Bay, Central Sound), representing eight sites (n = 94), thus, only regional-scale results will be presented here. (Fig 2). Brooded larvae collected from the ninth site, Little Skookum Inlet, were not included in the regional classification because they could be retained in their own region (South Sound); however, with few (n = 5) samples collected from this site and region, the classification algorithms could not successfully classify samples from this site.

Regional LDA classification success was optimized with eight elemental variables (Al:Ca, P:Ca, Mn:Ca, Fe:Ca, Zn:Ca, Sr:Ca, Ba:Ca, U:Ca). Classification to Central Sound was the most accurate (79.6%), followed by North Sound (69.7%) (Table 1). Classification accuracy was lowest when classifying Sequim Bay (66.7%) (Table 1). Overall, regional classification success was relatively high (74.5%). Linear discriminant analysis successfully classified *O. lurida* brooded larvae into three regions with 71.8 ± 0.004 (bootstrap sampling with 5000 permutations, 95% CI) accuracy. The bootstrap assignment accuracy of 71.8% is less than the jackknife classification assignment accuracy of 74.5% but provides improved accuracy because the bootstrap sampling tests a larger proportion of the data repeatedly. The first canonical function explained 67.95% of the total variance with strong positive loadings from Sr:Ca and Fe:Ca, and strong negative loadings by Mn:Ca and Zn:Ca (Table C, Supplemental Fig S1). The second canonical function explained the remaining 32.05% of the variance with strong positive loading of U:Ca and strong negative loading from Fe:Ca and Sr:Ca (Table C, Supplemental

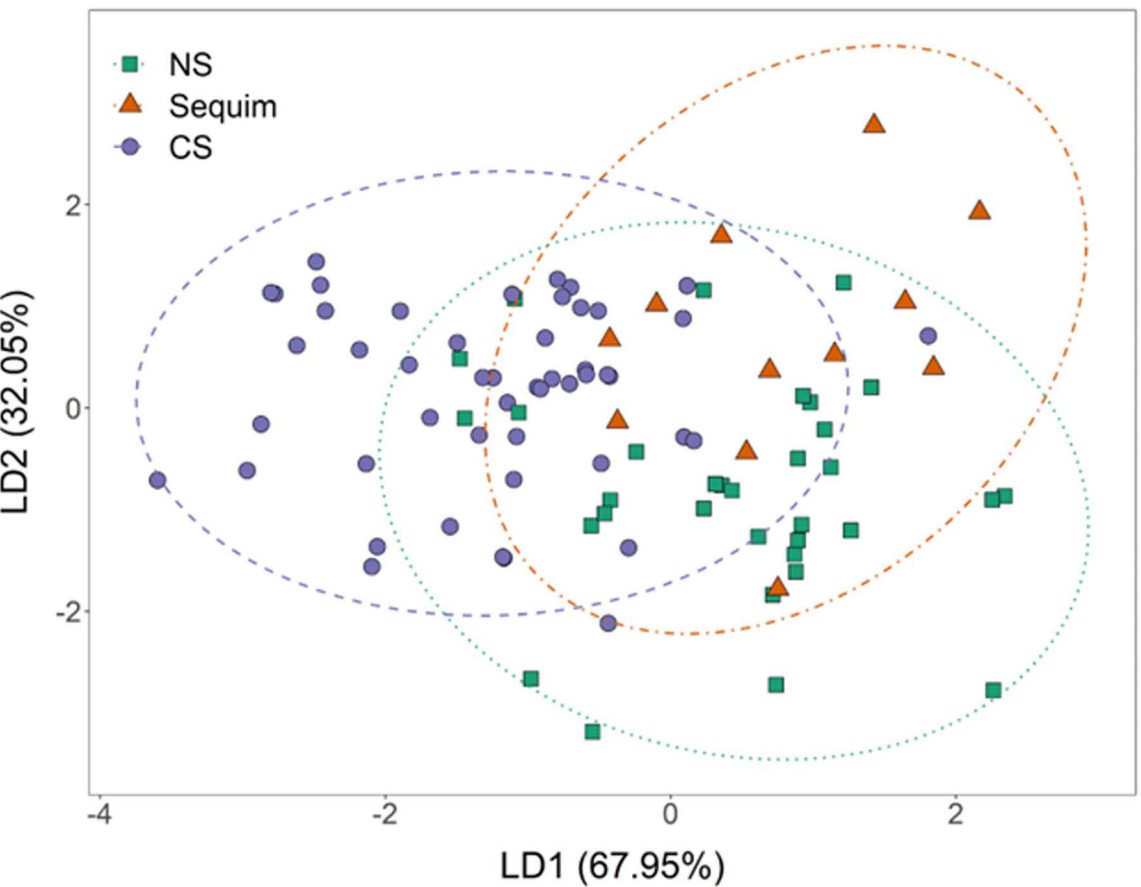

**Fig 2. Canonical score plot of linear discriminant function analysis conducted on *O. lurida* brood shell signatures among regions.** NS - North Sound, Sequim - Sequim Bay, CS - Central Sound. Each region is represented by a different shade and symbol combination, and ellipses represent 95% confidence intervals.

**Table 1. Confusion matrix of LDA analysis of *Ostrea lurida* larvae samples classified to region. The rows represent the actual region of the larval sample and the columns show the number of larval samples predicted to each region and the percentage that were predicted to their actual region of origin correctly. Regions are abbreviated: NS, North Sound; CS, Central Sound; and Seq, Sequim.**

| | | Predicted Group | | | |
|---|---|---|---|---|---|
| Actual Group | n | CS | NS | Seq | % correct |
| **LDA** | | | | | |
| Central Sound | 49 | 39 | 7 | 3 | 79.6% |
| North Sound | 33 | 4 | 23 | 6 | 69.7% |
| Sequim | 12 | 2 | 2 | 8 | 66.7% |

Fig S1). Central Sound and North Sound (77.1 ± 0.005 and 66.8 ± 0.008 assignment accuracy respectively, bootstrap sampling with 5000 permutations, 95% CI) were differentiated primarily by the first canonical function driven by significant differences in Fe:Ca (ANOVA, $F_{(2, 91)}$ = 14.00, P < 0.001; Tukeys post hoc HSD test P < 0.001; Supplemental Fig S1), Sr:Ca (ANOVA, $F_{(2,91)}$ = 9.98, P < 0.001; Tukeys post hoc HSD test P < 0.001; Supplemental Fig S1), and Al:Ca (ANOVA, $F_{(2,91)}$ = 8.67, P < 0.001; Tukeys post hoc HSD test P < 0.001; Supplemental Fig S1).

Sequim Bay (64.8 ± 0.01 assignment accuracy, bootstrap sampling with 5000 permutations, 95% CI) was distinguished from North and Central Sound with significant differences in Mn:Ca (ANOVA, $F_{(2,91)}$ = 4.79, P = 0.011; Tukeys post hoc HSD test, Seq-NS P = 0.049 and Seq-CS P = 0.007; Supplemental [Fig S1]). Sequim Bay was more similar to North Sound than Central Sound, only differing significantly in Mn:Ca, whereas Sequim and Central Sound differed significantly in two additional elements (Zn:Ca and Sr:Ca, Supplemental [Fig S1]). Although U:Ca was largely responsible for discrimination along the second canonical function, there were no significant differences among the regions when U:Ca was examined independently (ANOVA, $F_{(2,91)}$ = 2.07, P = 0.132).

## Temporal variation

The NMDS analysis of elemental ratios of brooded larvae collected from Fidalgo Bay and Dyes Inlet sampled repeatedly throughout the season produced convergence with low stress with weak ties ([Fig 3]). No within-season temporal differences in elemental signatures were detected between the early (6/15/2015–7/5/2017) and late (7/5/2015–8/15/2015) seasons at Dyes Inlet (ANOSIM, P = 0.48). In Fidalgo Bay, a small but significant difference was detected between early- and late-season broods (ANOSIM, R-statistic = 0.025, P = 0.006).

## Influence of non-lethal sampling method on signatures

The NMDS analysis of brooded larvae collected under the two sampling methods, $MgSO_4$ anesthetic and lethal collection, produced convergence with low stress with weak ties (Stress = 0.1995, $R^2$ = 0.92). Visual assessment of the ordination plots suggests the collection method to be a factor influencing sample scores. The results of ANOSIM show the two collection methods were significantly different but with a low R statistic indicating the two methods are

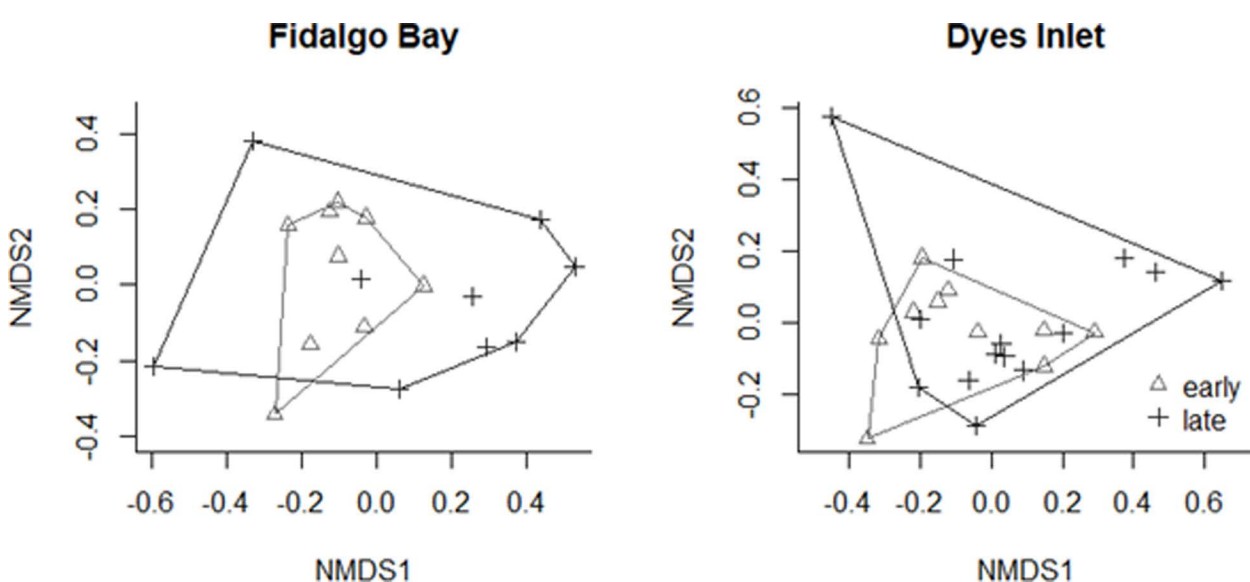

**Fig 3. NMDS ordination of elemental concentrations in brooded *O. lurida* larval shells compared for temporal variation between early (6/15/2015-7/5/2015) and late (7/5/2015-8/15/2015) reproductive season for two locations, Fidalgo Bay and Dyes Inlet.** Brooded larvae collected early in the season are depicted by triangles and those collected late in the season by plus signs.

not very different (R = 0.1125, significance (P) =.0255). The difference in elemental signatures was statistically significant, but the magnitude of the difference is not ecologically relevant. Elemental signatures of brooded larvae collected using anesthetized and lethal collection methods fell within brood signature LDA ordination space, although these samples were collected in North Sound during 2016 and plotted within the 95% confidence ellipse for Central Sound brood signatures collected in 2015 (Supplemental Fig S2).

## Assignment of settler signatures to brood origins

Core distributions of settler concentrations of individual elements displayed limited overlap with brood regions (Supplemental Fig S1). When plotted in brood ordination space, *O. lurida* settler elemental signatures (n = 154) were not fully encompassed by the LDA-defined brood regions and the elemental signatures from the two settler collection sites (Fidalgo Bay and Dyes Inlet) generally overlapped with one another and were not distinct (Fig 4). Broadly, the settler elemental signatures were more variable than the brood elemental signatures, and a substantial proportion of individual settler signatures (44%) fell outside of the 95% confidence ellipses for brood signatures. Additionally, 85% of settlers were estimated to have a >95% probability of originating outside the characterized brood sampling area based on combined Mn:Ca and Sr:Ca signatures. Together, these results suggest either broad regions of unsampled brood sources or a mismatch in analytical elemental sampling methodology between brooded larvae and settlers.

## Settler signature cluster analysis

MCMC mixture analysis was conducted on elemental signatures of *O. lurida* settlers collected in Fidalgo Bay and Dyes Inlet combined (n = 153), and model selection identified the presence of one cluster contributing to the settler cohort (Table D in S1 Appendix). The selection of a single putative source cluster may suggest a single geochemically defined brood origin for all settlers, but it is also plausible that geochemical differences among geographically distinct brood sources were not large enough to be detected using the MCMC approach.

## Discussion

This study explored the temporal and spatial scale of trace elemental signatures in the larval shells of a marine invertebrate in a glacially formed inland sea. We were able to distinguish populations from three Puget Sound regions with 75% accuracy using the chemical composition of brooded larvae collected at eight sites. This result is similar to that of Carson [17], who also could distinguish regional signatures in *O. lurida* but not sites. This is also a common result in other bivalve studies [19,20,21]. Although shell chemistry did not vary to the degree originally hypothesized, Puget Sound does apparently have sufficient variability to support further exploration of trace elemental fingerprinting studies. If similar to that of other calcified structures of marine organisms (e.g., larval shells, statoliths, otoliths), this suggests that ecological or restoration questions of population connectivity within Puget Sound and other inland seas can be potentially investigated among, but not within, sub-basins. The apparent stability of the signatures over a reproductive season also has important implications for the efficaciousness of this technique, as it would be logistically challenging to sample all potential source populations more than once a year. This apparent stability over a season is consistent with other systems [34] but can also be subject to abrupt shifts [44].

The three regions were distinguished based on the levels of eight elements. North Sound had the highest values of Fe and Zn, Central Sound was high in Ba and P, and Sequim was highest in U values. Sequim Bay was more similar to North Sound than Central Sound in

elemental composition, possibly because North Sound and Sequim are more directly influenced by the Pacific Ocean via the Strait of Juan de Fuca. The incorporation of elements into bivalve larval shells has been linked to various mechanisms, including temperature, salinity, watershed geology, and anthropogenic sources [34]. A full discussion of elemental incorporation in our system is beyond the scope of this manuscript. While the mechanisms driving differences in trace elemental uptake into calcified structures of molluscs remain largely unknown, this knowledge is not required to apply elemental fingerprinting as a technique for tracking natal origins.

Restoration practitioners attempting to create persistent populations of native species such as *O. lurida* can quantify recruitment to those populations without knowing the origin of those recruits. However, without larval tracking, they will not know whether those populations are primarily sustained through local reproduction or other sources. Genetic approaches to distinguishing populations of *O. lurida* suggest a single genetic group throughout inland waters of Washington State [45], where we distinguished three regions based on brooded shell chemistry. Understanding to what degree populations are sources or sinks allows for better spatial planning and evaluation of restoration efforts, including placing restoration

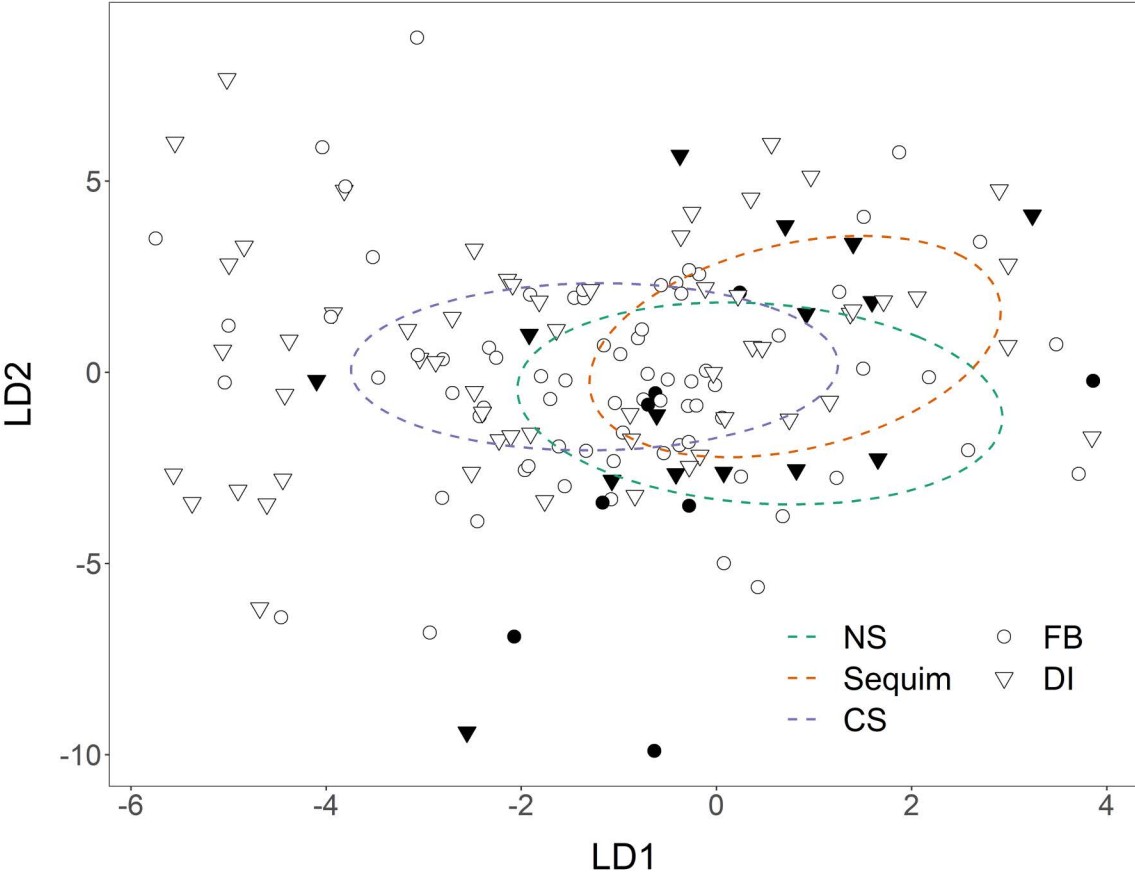

**Fig 4. *Ostrea lurida* settler elemental signature discriminant scores plotted in reference to linear discriminant analysis conducted on brood elemental signatures.** Of the 154 settlers analyzed, 85% (open symbols) were estimated to have a > 95% probability of originating from outside the characterized brood regions, with only 15% (closed symbols) deemed likely to be sourced to one of the characterized brood regions. Shapes represent the region of settler collection (FB - Fidalgo Bay, DI - Dyes Inlet). Confidence ellipses (95%) represent brood LDA geographic regions, and are identical to the ellipses in Fig 2.

sites relative to one another. The regional scale of variability in shell chemistry reported here implies that managers cannot distinguish settlers from various restoration sites within a subbasin, but would potentially be able to distinguish exchange among regions. Regional exchange has implications for determining a restored population's ability to adapt to local conditions or become genetically isolated. Regional-scale connectivity studies can still inform restoration planning [46] as managers consider factors such as resilience to localized catastrophes and the appropriate collection of broodstock used in creating seeds for new sites.

We could not confidently assign settled oysters back to putative source populations due to chemical differences between the settler samples and any of the three potential source regions defined by brood shell chemistry. Regardless of the collection site, the portion of the settler's shell formed during brooding (collected post-dispersal) was over 2x higher in mean Mn:Ca and Fe:Ca, whereas the brooded larval shell (collected pre-dispersal) had over 3x higher P:Ca and Ba:Ca (Supplemental Fig S1), and settlers occupied a more variable area of multivariate space relative to brood samples (Fig 4). Although it is theoretically possible that a large proportion of collected settlers originated in an unknown source population with distinct shell chemistry, it is more likely that methodological artifacts led to a shift in brooded shell chemistry analyzed from settlers. Despite efforts to sample from the same region of the shell for both brood and settler individuals, reflecting elemental signatures from shell material accreted while in brooding locales, shell material collection methodology differed by necessity as described below.

A key methodological difference in this study, compared to others that were able to assign settlers, was the decision to crush and analyze brooded larval shells in bulk. Individual analysis of ~ 150 um diameter shells did not yield sufficient counts-per-second of the elements of interest, and so the bulk analysis was needed to increase the amount of material analyzed and confidently characterize the shell chemistry of an overall brood. After settlement, however, the brooded shell cannot be separated from the shell formed in the plankton or after settlement in settlers; analysis of individual settlers was necessary. Although both analyses used LA-ICP-MS, the differences between shell preparation, amount of shell material available, and sampling may have influenced the results in comparison. Another possible issue with analyzing the larval shell on a settled bivalve is the potential laser burn-through to the non-target shell beneath the larval shell [47].

Because of the apparent lack of agreement between brood and settler signatures, we attempted to further explore the settler signatures in isolation using an MCMC clustering algorithm to estimate the number of geochemical brood source contributions to the settler cohort. This cluster analysis did not result in any distinct groupings despite the samples being collected from two widely separated locations that might be expected to have distinct larval source populations and despite demonstrating distinct source region geochemistry in this study. An unsupervised clustering approach such as MCMC is a valuable exploratory tool when prior information about sources is unavailable but is less reliable than assignment methods that incorporate training data from source material (e.g., LDA, MLE) [41]. Because Olympia oyster connectivity has been successfully evaluated using shell chemistry in a previous study [17], and brood source sampling is accessible in Puget Sound, further efforts in the region should focus on resolving the methodological mismatch between brood and settler LA-ICP-MS so that more robust LDA and MLE source assignment methods can be applied with confidence.

One new tool utilized in this study, removing brooded larvae using a relaxation agent to prevent mortality in adult oysters [31], did not apparently account for the chemical differences observed when compared to broods collected lethally (without the agent). Especially since all shells should have been already formed at the time of collection, proper cleaning of the

larval shells should allow this method to be used in future studies. The benefit of an increased sample size to search for broods, together with lower mortality of animals at a restoration site, makes the relaxation technique worthwhile despite the use of $MgSO_4$ on samples that will later be analyzed for trace elemental composition.

Becker et al. [48] observed that the peak abundance of brooding, planktonic larvae, and settlement in 2013 peaked sequentially at one restored population of *O. lurida*, taken as observational evidence for a high degree of self-recruitment there. During sample collection for this study, we again made observations consistent with significant local retention of larvae and a two-week pelagic larval duration. Brooding in restored populations in both Fidalgo Bay [49] and Dyes Inlet peaked about one week prior to peak abundance of planktonic larvae in nearby waters, and two weeks before peak settlement at each site. Although our goal of more definitively identifying the source of these settlers was not realized, a finding of a high degree of self-recruitment within subbasins would be consistent with evidence of local adaption and modest genetic isolation-by-distance [6]. If recruits to these restored populations are indeed mostly local-origin, this bodes well for the demographic sustainability of those populations but also may suggest that new populations are unlikely to establish naturally. Future investigation will hopefully be able to make methodological improvements to track population connectivity and better inform the restoration of the West Coast's only native oyster.

## Supporting information

**S1 Appendix. Supplemental materials including methods, figures, and tables.** (DOCX)

**S1 Dataset. All raw data required to replicate the study.** (XLSX)

**S1 Fig. Elemental ratios to calcium for the 8 elements used to classify brooded larval shells of *O. lurida* with linear discriminant analysis (LDA) from the three regions North Sound (NS), Sequim Bay (Sequim) and Central Sound (CS) in Puget Sound, WA.** Horizontal lines represent median values; lower and upper hinges represent the 25th and 75th percentiles, respectively; whiskers extend to the largest and smallest measured value within 1.5 x interquartile range (IQR; difference between 75th and 25th percentile); filled circles represent outliers beyond 1.5 x IQR (some upper limits truncated for ease of visualization). The results of individual ANOVAs (df = 2) are shown as p value and different letters above bars indicate significant differences (p < 0.05) from Tukey post hoc tests. Elemental ratios of settled recruits (Set) of unknown brood origin are plotted here (unshaded boxes) for comparison but were not included in the ANOVA. (TIF)

**S2 Fig. NMDS ordination of elemental concentrations in brooded larval shells collected using non-lethal $MgSO_4$ anesthetic and lethal collection.** Left plot analysis are labeled to sample collection method and the right plot analysis are labeled according to the Olympia oyster the larvae were collected from. (TIF)

## Acknowledgments

The authors wish to thank B. Peabody at the Puget Sound Restoration Fund, B. Vadopalas at the University of Washington Seattle, R. Hunter and J. Emm at the Northwest Indian College, B. Sizemore at the Washington Department of Fish and Wildlife, and A. Bullock, J. Gonzaga, M. McCartha, H. Parker and many other students at the University of Washington Tacoma.

## Author contributions

**Conceptualization:** Megan Hintz, Bonnie J Becker, Henry S. Carson, Marco B.A. Hatch, Brian Allen.

**Data curation:** Megan Hintz.

**Formal analysis:** Megan Hintz, Verena H. Wang, Brian Rusk.

**Funding acquisition:** Bonnie J Becker, Henry S. Carson, Marco B.A. Hatch, Brian Allen.

**Investigation:** Megan Hintz, Bonnie J Becker, Henry S. Carson, Marco B.A. Hatch, Brian Allen.

**Methodology:** Megan Hintz, Bonnie J Becker, Henry S. Carson, Marco B.A. Hatch, Brian Allen, Brian Rusk.

**Project administration:** Megan Hintz, Bonnie J Becker.

**Supervision:** Bonnie J Becker.

**Validation:** Megan Hintz, Verena H. Wang.

**Visualization:** Megan Hintz, Henry S. Carson, Verena H. Wang.

**Writing – original draft:** Megan Hintz, Bonnie J Becker, Henry S. Carson, Verena H. Wang.

**Writing – review & editing:** Megan Hintz, Bonnie J Becker, Henry S. Carson, Verena H. Wang.

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
