## [Decision Letter · Decision Letter 0]

16 Sep 2024

PONE-D-24-06389Larval shell chemistry of the Olympia Oyster (Ostrea lurida) in Puget Sound, WA to assess population connectivity and restoration planningPLOS ONE

Dear Dr. Becker,

Thank you for submitting your manuscript to PLOS ONE. After careful consideration, we feel that it has merit but does not fully meet PLOS ONE’s publication criteria as it currently stands. Therefore, we invite you to submit a revised version of the manuscript that addresses the points raised during the review process.

We look forward to receiving your revised manuscript.

Kind regards,

Vanessa Carels

Staff Editor

PLOS ONE

3. Please ensure that you include a title page within your main document. We do appreciate that you have a title page document uploaded as a separate file, however, as per our author guidelines (http://journals.plos.org/plosone/s/submission-guidelines#loc-title-page) we do require this to be part of the manuscript file itself and not uploaded separately.

“BJB, Washington Sea Grant (R/HCE-8), https://wsg.washington.edu/.”

“This study was funded by Washington Sea Grant (R/HCE-8). The authors wish to thank B. Peabody at the Puget Sound Restoration Fund, B. Vadopalas at the University of Washington Seattle, R. Hunter and J. Emm at the Northwest Indian College, B. Sizemore at the Washington Department of Fish and Wildlife, and A. Bullock, J. Gonzaga, M. McCartha, H. Parker and many other students at the University of Washington Tacoma.”

“BJB, Washington Sea Grant (R/HCE-8), https://wsg.washington.edu/.”

6. We note that your Data Availability Statement is currently as follows: [All relevant data are within the manuscript and its Supporting Information files.]

7. Please ensure that you include a title page within your main document. You should list all authors and all affiliations as per our author instructions and clearly indicate the corresponding author.

8. We note that you have included the phrase “data not shown” in your manuscript. Unfortunately, this does not meet our data sharing requirements. PLOS does not permit references to inaccessible data. We require that authors provide all relevant data within the paper, Supporting Information files, or in an acceptable, public repository. Please add a citation to support this phrase or upload the data that corresponds with these findings to a stable repository (such as Figshare or Dryad) and provide and URLs, DOIs, or accession numbers that may be used to access these data. Or, if the data are not a core part of the research being presented in your study, we ask that you remove the phrase that refers to these data.

9. We note that Figure 1 in your submission contain [map/satellite] images which may be copyrighted. All PLOS content is published under the Creative Commons Attribution License (CC BY 4.0), which means that the manuscript, images, and Supporting Information files will be freely available online, and any third party is permitted to access, download, copy, distribute, and use these materials in any way, even commercially, with proper attribution. For these reasons, we cannot publish previously copyrighted maps or satellite images created using proprietary data, such as Google software (Google Maps, Street View, and Earth). For more information, see our copyright guidelines: http://journals.plos.org/plosone/s/licenses-and-copyright.

We require you to either (1) present written permission from the copyright holder to publish these figures specifically under the CC BY 4.0 license, or (2) remove the figures from your submission.

10. Please include captions for your Supporting Information files at the end of your manuscript, and update any in-text citations to match accordingly. Please see our Supporting Information guidelines for more information: http://journals.plos.org/plosone/s/supporting-information.

Reviewers' comments:

Reviewer's Responses to Questions

**Comments to the Author**

1. Is the manuscript technically sound, and do the data support the conclusions?

Reviewer #1: Partly

Reviewer #2: Yes

2. Has the statistical analysis been performed appropriately and rigorously? 

Reviewer #1: Yes

Reviewer #2: Yes

3. Have the authors made all data underlying the findings in their manuscript fully available?

Reviewer #1: No

Reviewer #2: Yes

4. Is the manuscript presented in an intelligible fashion and written in standard English?

Reviewer #1: Yes

Reviewer #2: Yes

5. Review Comments to the Author

Reviewer #1: This manuscript describes the elemental shell chemistry of the earliest shell formed by Olympia oysters, which is a method to determine population connectivity. The collection of field samples and LA-ICP-MS methods are well described and seem appropriate to the question. The authors are able to compare lethal vs non-lethal collection of brooded larvae. They also are up front about how the crushing or laser ablation of samples may have contributed to the inability to link settling oysters to regional brood sources, while addressing biological explanations as well. My only big suggestion is to clarify whether the settlers were different from the brood sources AND different from each other (that is, more variable overall than the brood sources, as it appears in Fig. 4), or alternatively the settlers cluster together but not with any of the brood sources. Similarly, in Fig. 3 and in the data analysis section, be specific about whether the recruits from Dyes and Fidalgo were analyzed separately or together (and if together, that these two large regions were not distinguishable at this life stage, given the laser ablation technique).

For the underlying data, it was not apparent to me that the actual elemental values were available for 99 brooded larvae from 9 sites and the >100 recruits from 2 sites. Perhaps I missed those tables in the supplement, but an online spreadsheet would be an alternative, for instance in Mendeley Data.

Specific line comments:

Line 41: needs citation, e.g. Ridlon et al. 2021 (10.1007/s12237-021-00920-7)

Line 164: what is petrographic? Maybe line 169 needs to come earlier

Line 188: LOD?

Line 202: not Ca?

Line 211: 9 variables sounds like these might be raw elemental concentrations rather than standardized to Ca? But it would be helpful to be specific

Line 251: specify which response variables were included in this multivariate analysis comparing early and late season at 2 sites

Line 254: you define ANOSIM slightly differently here relative to above

Line 278: is this a mismatch in how the shell is sampled? Or in not having the necessary spatial coverage of sites? (line 373 is more clear in this regard)

Line 341: Can you specifically state whether the recruits were analyzed together from Dyes and Fidalgo or separately? The panels of Fig. 3 show the sites separately, and the statistics only address the sites separately. But I think readers would be very interested to know if the settlers from Dyes and Fidalgo overlapped completely in their elemental signatures. (Fig. 4 makes it appear that the recruits were more variable in elemental signatures than were the brooded larvae, but again there is no coding by recruitment site, so not possible to infer.)

Line 387: is it worth stating that the recruits were more variable in elemental signatures than were the brooded larvae (rather than representing a tight grouping that was distinct from the brooded larvae)?

Line 418: perhaps a place to site Silliman 2018 Evolutionary Applications with a statement such as: Genetic approaches to distinguishing populations of O. lurida suggest a single genetic group throughout inland waters of Washington State (Silliman 2018), where we distinguished three regions based on brooded shell chemistry.

Line 430-432: this content seems important but I do not know how to compare these statements to any of the data you visualize earlier. Also, what is the difference between a brooded portion of the shell and a pre-dispersal portion of the shell: these seem like they should be the same? Also another place to comment on whether the samples from recruits were a) more variable than the brooded regions or b) clustered in a distinct portion of multivariate space from the brooded regions.

Reviewer #2: Interesting study, methods and analysis well done. I'd suggest adding, if possible, some information on the potential dispersal distance and time the veligers have in the water, I suspect their potential dispersal distance is shorter than oysters that do not brood but I'm not sure.

6. PLOS authors have the option to publish the peer review history of their article (what does this mean? ). If published, this will include your full peer review and any attached files.

**Do you want your identity to be public for this peer review?** For information about this choice, including consent withdrawal, please see our Privacy Policy .

Reviewer #1: **Yes: ** Jennifer L Ruesink

Reviewer #2: **Yes: ** David M Schulte

---

## [Author Response · Author response to Decision Letter 1]

31 Oct 2024

(This is a copy of the text in the Response for Reviewers file.)

We would like to express our gratitude to you and the reviewers for the constructive feedback and valuable insights provided during the review of our manuscript, "Larval shell chemistry of the Olympia Oyster (Ostrea lurida) in Puget Sound, WA to assess population connectivity and restoration planning." We appreciate the time and effort taken to assess our work, and we are grateful for the opportunity to revise and resubmit our paper.

In response to the reviewers' comments, we have made all suggested changes to enhance the clarity of our manuscript and consistency with PLOS ONE guidelines. Below, we provide a detailed account of how we have addressed each specific comment raised during the review process. We believe these revisions have strengthened our paper and are pleased to share the updated version with you.

Specifically, we addressed the following comments:

Academic Editor

1. Please ensure that your manuscript meets PLOS ONE's style requirements.

Author Response: All necessary adjustments have been made to align with the journal's formatting guidelines.

2. Please provide additional information regarding the permits you obtained for the work.

Author Response: The work was conducted with a scientific collection permit authorized by the Washington Department of Fish and Wildlife and permission by private land owners on private land. We revised the text to include the following text in lines 117-120 (formerly 102-105):

"All field collections for O. lurida adults, brooded larvae, and later for post-set recruits on public lands were authorized by a Scientific Collection Permit issued by the Washington Department of Fish & Wildlife; on private lands, permissions for access and collection were provided by the owner."

3. Please ensure you include a title page within your main document.

Author Response: This adjustment was made.

4. Please state what role the funders took in the study.

Author Response: Here is our amended Role of Funder statement. Thank you for changing the online submission on our behalf.

This study was funded by Washington Sea Grant (R/HCE-8, BJB), https://wsg.washington.edu/. The funders had no role in study design, data collection and analysis, decision to publish, or preparation of the manuscript.

5. Please remove any funding related text from the manuscript and let us know how you would like to update your funding statement.

Author Response: Text removed (line 512, formerly on line 483). The updated funding statement is provided in comment 4 above.

6. Please confirm at this time whether or not your submission contains all raw data required to replicate the results of your study.

Author Response: We have included all of our raw data in a supplemental file, S2 Dataset. A caption has been added to our original manuscript (lines 659-661).

7. Please ensure that you include a title page within your main document.

Author Response: This adjustment was made.

8. We note that you have included the phrase "data not shown" in your manuscript. Unfortunately, this does not meet our data sharing requirements.

Author Response: We removed the reference to "data not shown" (paragraph starting on line 201, formerly on line 192).

9. We note that Figure 1 in your submission contains map images which may be copyrighted.

Author Response: The map in Figure 1 was created in R programming and contains no copyrighted images. We added a reference to the PBSmapping package used to create the map (lines 130-131).

10. Please include captions for your supporting information files at the end of your manuscript, and update any in-text citations to match accordingly.

Author Response: We have now relabeled our original supplemental file as S1 Appendix and all of our raw data as S2 Dataset. Captions for both are at the end of our manuscript.

Reviewer 1

Overall: This manuscript describes the elemental shell chemistry of the earliest shell formed by Olympia oysters, which is a method to determine population connectivity. The collection of field samples and LA-ICP-MS methods are well described and seem appropriate to the question. The authors are able to compare lethal vs non-lethal collection of brooded larvae. They also are up front about how the crushing or laser ablation of samples may have contributed to the inability to link settling oysters to regional brood sources, while addressing biological explanations as well. My only big suggestion is to clarify whether the settlers were different from the brood sources AND different from each other (that is, more variable overall than the brood sources, as it appears in Fig. 4), or alternatively the settlers cluster together but not with any of the brood sources. Similarly, in Fig. 3 and in the data analysis section, be specific about whether the recruits from Dyes and Fidalgo were analyzed separately or together (and if together, that these two large regions were not distinguishable at this life stage, given the laser ablation technique).

For the underlying data, it was not apparent to me that the actual elemental values were available for 99 brooded larvae from 9 sites and the >100 recruits from 2 sites. Perhaps I missed those tables in the supplement, but an online spreadsheet would be an alternative, for instance in Mendeley Data.

Author Response: We would like to thank Review 1 for their thorough review of the manuscript. We agree with their suggestion that improved clarification about the variability in brood and settler samples would benefit the manuscript and we have addressed that in the responses below. Additionally, we clarified in the methods that samples from the two sites were analyzed separately (line 275, formerly 253) for temporal variation. We have included all of the raw data in the supplementary information in S2 Dataset.

1. Line 41: needs citation, e.g. Ridlon et al. 2021 (10.1007/s12237-021-00920-7)

Author Response: Thank you for this suggestion. We have added a reference to Ridlon et al. 2021 in line 57 (formerly line 41).

2. Line 164: what is petrographic? Maybe line 169 needs to come earlier

Author Response: A "petrographic glass slide" is the specific type of glass slide that was used in this study. However, the kind of slide had no impact on the study design, so all references to "petrographic slide" and "petrographic glass slide" were removed and replaced with "glass slide" to avoid any confusion (now lines 174, 175, and 181).

3. Line 188: LOD?

Author Response: Thank you for pointing out the use of an acronym without a definition. We've updated lines 206-207 (formerly 195) to include limits of detection (LOD).

4. Line 202: not Ca?

Author Response: We appreciate the reviewer pointing out this confusion and have provided further clarification. All trace elements were analyzed as relative ratios to calcium, and we added "relative ratios to 42Ca" in line 221 to clarify.

5. Line 211: 9 variables sounds like these might be raw elemental concentrations rather than standardized to Ca? But it would be helpful to be specific

Author Response: We appreciate the reviewer pointing out this confusion and have provided further clarification. All elemental values were standardized to Ca. We added clarification to line 223 (formerly line 211), correcting "trace elements" to state "trace elemental ratios."

6. Line 251: specify which response variables were included in this multivariate analysis comparing early and late season at 2 sites

Author Response: All elemental ratios collected to create elemental signatures were used in the temporal variation analysis. The text "All nine elemental ratios (Mg:Ca, Al:Ca, P:Ca, Mn:Ca, Fe:Ca, Zn:Ca, Sr:Ca, Ba:Ca, U:Ca) were used in this multivariate analysis." was added to line 270 to clarify.

7. Line 254: you define ANOSIM slightly differently here relative to above

Author Response: We agree with the reviewer that this is confusing for the reader. We corrected this by describing the methods clearly in the "temporal variation" subsection (line 360) and now reference that subsection in the following subsection "Influence of non-lethal sampling method on signatures" (line 372). This included moving some of the method description text up (see track changes).

8. Line 278: is this a mismatch in how the shell is sampled? Or in not having the necessary spatial coverage of sites? (line 373 is more clear in this regard)

Author Response: We agree with the reviewer that (formerly) line 373 was more clear about how the mismatch could be due to a mismatch in methodology or not having the spatial coverage of sites, and we edited lines 297-298 to clarify and match by including "or the contribution of unsampled brood origin sites."

9. Line 341: Can you specifically state whether the recruits were analyzed together from Dyes and Fidalgo or separately? The panels of Fig. 3 show the sites separately, and the statistics only address the sites separately. But I think readers would be very interested to know if the settlers from Dyes and Fidalgo overlapped completely in their elemental signatures. (Fig. 4 makes it appear that the recruits were more variable in elemental signatures than were the brooded larvae, but again there is no coding by recruitment site, so not possible to infer.)

Author Response: There was quite a bit of overlap between both settler collection sites, and recruits were indeed more variable in their signatures than the brooded larvae. To communicate this to the reader we updated figure 4 to include labeling of recruitment sites and added the following text was added to line 388-389, "and the elemental signatures from the two settler collection sites (Fidalgo Bay and Dyes Inlet) generally overlapped with one another and were not distinct (Fig 4)."

10. Line 387: is it worth stating that the recruits were more variable in elemental signatures than were the brooded larvae (rather than representing a tight grouping that was distinct from the brooded larvae)?

Author Response: We agree with the reviewer that clearly stating that recruiters were more variable would be valuable to the reader. In response, we added "Broadly, the settler elemental signatures were more variable than the brood elemental signatures…" to line 389.

11. Line 418: perhaps a place to site Silliman 2018 Evolutionary Applications with a statement such as: Genetic approaches to distinguishing populations of O. lurida suggest a single genetic group throughout inland waters of Washington State (Silliman 2018), where we distinguished three regions based on brooded shell chemistry.

Author Response: We appreciate the reviewer suggesting adding the local genetic work on O. luria in Washington and have added the suggested text to lines 441-444.

12. Line 430-432: this content seems important but I do not know how to compare these statements to any of the data you visualize earlier. Also, what is the difference between a brooded portion of the shell and a pre-dispersal portion of the shell: these seem like they should be the same? Also another place to comment on whether the samples from recruits were a) more variable than the brooded regions or b) clustered in a distinct portion of multivariate space from the brooded regions.

Author Response: There is no difference between the brooded portion of the shell and the pre-dispersal portion of the shell; they are the same. We provided clarification in the text including a reference to the supplemental files and included another comment about the variability of signatures in brood and settler samples in the paragraph starting at line 453). The updated text states: "Regardless of the collection site, the portion of the settler's shell formed during brooding (collected post-dispersal) was over 2x higher in mean Mn:Ca and Fe:Ca, whereas the pre-dispersal brooded larval shell (collected pre-dispersal) had over 3x higher P:Ca and Ba:Ca (Figure A in S1 Appendix), and settlers occupied a more variable are of multivariate space relative to brood samples (Fig 4)."

Reviewer 2

Interesting study, methods and analysis well done. I'd suggest adding, if possible, some information on the potential dispersal distance and time the veligers have in the water, I suspect their potential dispersal distance is shorter than oysters that do not brood but I'm not sure.

Author Response: We would like to thank Reviewer 2 for their review and comment. We agree that adding context about the general larval dispersal distance potential would be beneficial and added the following language in lines 78-80, "Because their larvae are brooded for the first phase of development, the pelagic larval duration and dispersal potential of O. lurida is likely shorter than oysters that do not brood. The actual pelagic larval duration is unknown, but is as short as 7 days in laboratory trials [18]."

We hope these revisions are consistent with your expectations in preparing our manuscript for publication in PLOS ONE. Thank you for the opportunity to resubmit, and we look forward to your next decision.

---

## [Decision Letter · Decision Letter 1]

26 Nov 2024

PONE-D-24-06389R1Larval shell chemistry of the Olympia Oyster (Ostrea lurida) in Puget Sound, WA to assess population connectivity and restoration planningPLOS ONE

Dear Dr. Becker,

Thank you for submitting your manuscript to PLOS ONE. After careful consideration, we feel that it has merit but does not fully meet PLOS ONE’s publication criteria as it currently stands. Therefore, we invite you to submit a revised version of the manuscript that addresses the minor points raised during the review process.

We look forward to receiving your revised manuscript.

Kind regards,

Michael Schubert

Academic Editor

PLOS ONE

Journal Requirements:

Reviewers' comments:

Reviewer's Responses to Questions

**Comments to the Author**

1. If the authors have adequately addressed your comments raised in a previous round of review and you feel that this manuscript is now acceptable for publication, you may indicate that here to bypass the “Comments to the Author” section, enter your conflict of interest statement in the “Confidential to Editor” section, and submit your "Accept" recommendation.

Reviewer #1: All comments have been addressed

2. Is the manuscript technically sound, and do the data support the conclusions?

Reviewer #1: Yes

3. Has the statistical analysis been performed appropriately and rigorously? 

Reviewer #1: Yes

4. Have the authors made all data underlying the findings in their manuscript fully available?

Reviewer #1: Yes

5. Is the manuscript presented in an intelligible fashion and written in standard English?

Reviewer #1: Yes

6. Review Comments to the Author

Reviewer #1: I did catch two small issues in this revision, which overall reads really well and is very clear, thanks!

Line 126 says 8 locations, and line 142 says 9 where larvae were collected from brooding adults, line 230 says 8, and line 317 says 9, and also finally helps in understanding why 8 and 9 have been mentioned, since there must be 1 site that had only 5 samples. I think it would make sense to say 9 sites UNTIL you get to the analysis where you leave out your most southern site.

Line 458: area?

7. PLOS authors have the option to publish the peer review history of their article (what does this mean? ). If published, this will include your full peer review and any attached files.

**Do you want your identity to be public for this peer review?** For information about this choice, including consent withdrawal, please see our Privacy Policy .

Reviewer #1: **Yes: ** Jennifer Ruesink

---

## [Editor Report · Decision Letter 2]

14 Feb 2025

Larval shell chemistry of the Olympia Oyster (Ostrea lurida) in Puget Sound, WA to assess population connectivity and restoration planning

PONE-D-24-06389R2

Dear Dr. Becker,

We’re pleased to inform you that your manuscript has been judged scientifically suitable for publication and will be formally accepted for publication once it meets all outstanding technical requirements.

Kind regards,

Michael Schubert

Academic Editor

PLOS ONE

---

## [Editor Report · Acceptance letter]

PONE-D-24-06389R2

PLOS ONE

Dear Dr. Becker,

I'm pleased to inform you that your manuscript has been deemed suitable for publication in PLOS ONE. Congratulations! Your manuscript is now being handed over to our production team.

Kind regards,

on behalf of

Dr. Michael Schubert

Academic Editor

PLOS ONE